# Seasonal activities of the phyllosphere microbiome of perennial crops

Adina Howe[1,2,3], Nejc Stopnisek [4,5,6], Shane K. Dooley[1,2], Fan Yang[1], Keara L. Grady [4,5] & Ashley Shade [4,5,6,7,8,9] ✉

Understanding the interactions between plants and microorganisms can inform microbiome management to enhance crop productivity and resilience to stress. Here, we apply a genome-centric approach to identify ecologically important leaf microbiome members on replicated plots of field-grown switchgrass and miscanthus, and to quantify their activities over two growing seasons for switchgrass. We use metagenome and metatranscriptome sequencing and curate 40 medium- and high-quality metagenome-assembled-genomes (MAGs). We find that classes represented by these MAGs (Actinomycetia, Alpha- and Gamma- Proteobacteria, and Bacteroidota) are active in the late season, and upregulate transcripts for short-chain dehydrogenase, molybdopterin oxidoreductase, and polyketide cyclase. Stress-associated pathways are expressed for most MAGs, suggesting engagement with the host environment. We also detect seasonally activated biosynthetic pathways for terpenes and various non-ribosomal peptide pathways that are poorly annotated. Our findings support that leaf-associated bacterial populations are seasonally dynamic and responsive to host cues.

Perennial plants are a crucial target for the sustainable development of biofuels[1–3]. In addition to yielding high biomass that can be converted to biofuels and bioproducts, perennial crops offer a broad range of ecosystem services that support efforts to mediate climate change, including greenhouse gas mitigation and promotion of soil nutrient cycling[1,4–6]. Like all plants, perennials harbor diverse microbiota, and many of these microbes are either known or expected to benefit their hosts. For example, plant-associated microbes can increase productivity and protect against environmental stressors. Because of the intimate engagement of many plant-associated microbiome members with the host, management of the plant microbiome is one tool proposed to promote crop vigor and support crop resilience to global climate changes[7–10]. Therefore, along with selective breeding and data-informed field management, regulating the plant microbiome is expected to be strategic for the sustainable production of biofuel feedstocks.

Plants have anatomical compartments that each are inhabited by distinctive microbial consortia. Generally, the diversity and composition of the plant microbiome narrow from external compartments to internal, and the plant plays an active role in filtering the microbiome composition inward[11–13]. External plant compartments include the root zone, rhizosphere and rhizoplane below ground, and the epiphytic phyllosphere above ground[14]. External compartments have a relatively higher representation of transient or commensal microbial taxa, and these compartments engage with and recruit microbes from the immediate environment. Internal compartments include the

[1]Department of Agricultural and Biosystems Engineering, Iowa State University, Ames, IA 50011, USA. [2]Department of Bioinformatics and Computational Biology, Iowa State University, Ames, IA 50011, USA. [3]Center for Advanced Bioenergy and Bioproducts Innovation, Ames, IA 50011, USA. [4]The Great Lakes Bioenergy Research Center, Michigan State University, East Lansing, MI 48824, USA. [5]Department of Microbiology and Molecular Genetics, Michigan State University, East Lansing, MI 48824, USA. [6]The Plant Resilience Institute, Michigan State University, East Lansing, MI 48824, USA. [7]Department of Plant, Soil, and Microbial Sciences, Michigan State University, East Lansing, MI 48824, USA. [8]Program in Ecology, Evolution, and Behavior, Michigan State University, East Lansing, MI 48824, USA. [9]Present address: Univ Lyon, CNRS, INSA Lyon, Université Claude Bernard Lyon 1, Ecole Centrale de Lyon, Ampère UMR5005, 69134 Ecully cedex, France. ✉e-mail: ashley.shade@cnrs.fr

endosphere of above- and below-ground tissues, and these have relatively low richness and harbor the most selected microbiota[15,16]. Of these compartments, the rhizosphere has received the most attention as a critical site of microbial-plant interactions that are important for nutrient and water acquisition (e.g., Kuzyakov and Razavi[17]). However, members of the microbiota that inhabit the phyllosphere also provide important plant functions, such as pathogen exclusion and immune priming[18,19]. Phyllosphere microorganisms have specialized adaptations to their exposed lifestyle[16,20–22] and contribute to global carbon and other biogeochemical cycling, including transformations relevant to climate change[23–25], and inhabit the largest above-ground surface area[26]. Because perennial biofuel feedstocks often are selected to maximize foliage surface area, understanding the phyllosphere microbiome is expected to provide insights into microbial engagements that benefit the plant to support productivity and stress resilience.

There are two general challenges in regulating the microbiome to promote crop vigor and resilience to environmental stress. The first challenge is to distinguish the beneficial members of the plant microbiome from transient or commensal members, with the recognition that some members that provide host benefits likely change situationally, either over plant development or given environmental stress[27–29], while others are stable[30,31]. The second challenge is that plants and their agroecosystems are temporally dynamic over the growing season, and their associated microbiota is also dynamic. It is currently unclear what functions may be associated with phyllosphere microbial dynamics and their potential interactions with plant hosts.

Previously, we used 16S rRNA gene amplicon analysis to identify a "core" cohort of bacterial and archaeal taxa that were persistently associated with the phyllosphere microbiomes of two perennial biofuel feedstocks, miscanthus, and switchgrass. Persistent membership was established by collecting leaf samples over replicated field plots, over a temperate seasonal cycle, and across two annual growing seasons for switchgrass[32]. Other studies in switchgrass have similarly reported that the leaf can be distinguished from other plant compartments by its microbiome composition (e.g.,[33–37]), suggesting selection to the leaf compartment. In the current study, we aimed to understand the functional attributes and activities of persistent phyllosphere taxa, with an interest in specialized adaptations to the leaf and interactions with the host plant that may inform the mechanisms and nature of plant-microbe engagements. Therefore, we performed a seasonal analysis of phyllosphere metagenomes for miscanthus and switchgrass using nucleic acids from the same samples as used for amplicon analysis. We paired our metagenome longitudinal series with metatranscriptome analyses at select time points in switchgrass phenology to determine which functions were active seasonally. We performed genome-centric analyses of the metagenome and focused on understanding the seasonal dynamics and functions of a focal subset of medium- and high-quality metagenome-assembled-genomes (MAGs) that we could bin from these data. Our results reveal functions supportive of a leaf-associated lifestyle and seasonal activities of persistent phyllosphere members. Finally, we provide evidence that these genomes were detected in various sites and years beyond our original study plots, suggesting that they are general, consistent inhabitants of midwestern U.S. bioenergy grasses.

## Results

### Overview of results
Here, we advanced from our previous microbiome profiling (with 16S rRNA amplicon sequencing) to understand the temporal dynamics of the epiphytic phyllosphere microbiome of the perennial biofuel feedstocks miscanthus and switchgrass[32] to now investigate the functional genes and seasonal activities of the focal phyllosphere populations. In our previous study, we identified "core" members of these microbiomes based on their temporal persistence and occupancy.

Here, we assembled metagenome-assembled genomes to focus analysis on the most persistent and abundant populations and their seasonal dynamics, and then subsequently recruited metatranscripts to the focal MAGs to understand their seasonal transcript dynamics for switchgrass. We first report overlap between the two different datasets in the detection of "core" and "focal" populations and their taxonomy, then move to describe the general seasonal dynamics of MAGs and their seasonal transcript dynamics, and finally explore the consistent pathways detected and activated across focal MAGs that may be suggestive of a leaf-associated and/or host-responsive lifestyle. Also, we searched for these focal MAGs in other mid-western metagenomes to understand more about their distributions.

We used a MAG-based approach (as opposed to isolation and genome sequencing at each time point) for three main reasons. First and importantly, we used a MAG-based approach so that we could capture yet-uncultivable members and query their functional genes without imposing cultivation biases in our understanding of the phyllosphere microbiome. Second, we wanted to achieve a match of observed transcripts to functional genes from the same pool of cell lyses and directly paired RNA and DNA co-extracts. Thirdly, we wanted to leverage the throughput of cultivation-independent sequencing to achieve relatively higher resolution over time and space than what is possible given the same cost and effort using cultivation-dependent genome sequencing of isolates.

Switchgrass and miscanthus leaves were collected at the Great Lakes Bioenergy Research Center (GLBRC) located at the Kellogg Biological Station (KBS) in Hickory Corners, MI, USA (42º23'41.6" N, 85º22'23.1" W) (Fig. 1). We sampled switchgrass (*Panicum virgatum L.* cultivar Cave-in-rock) and miscanthus (*Miscanthus x giganteus*) from the Biofuel Cropping System Experiment (BCSE) sites (plot replicates 1–4,[32]. We collected leaves from switchgrass and miscanthus at eight and nine time points, respectively, in 2016 and switchgrass at seven time points in the 2017 season (Table 1, Fig. 1, Supplementary Data 1, Supplementary Data 2). From the leaf surfaces, we isolated nucleic acids for metagenome and metatranscriptome analysis (Fig. 2), which included quality control and filtering (Fig. S1, Table S1), assembly, mapping, annotation, genome binning, and dereplication (see Methods).

### Focal leaf bacterial populations and general dynamics
Among all assembled MAGs ($n = 238$), we focused analysis on 40 that were high- and medium-quality based on completeness and contamination standards (Fig. 3A, Supplementary Data 3)[38]. The consistent detection of these 40 MAGs in both miscanthus and switchgrass samples (Fig. 4) suggests that their originating populations were not host-specific but rather distributed among these perennial grasses; this supports our previous results from the 16S rRNA gene amplicon survey that revealed substantial overlap in the major leaf bacterial taxa across the two crops[32]. Furthermore, there were 38 "core" OTUs from our prior survey that were also found to be associated with 25 focal MAGs per their taxonomic identifications (Fig. 3B); our previous study prioritized OTUs based on abundance and seasonal and field-replicated occupancy, and so this conservatively suggests that more than half of the focal MAGs represent the most abundant and consistently detected taxa in this ecosystem. The 40 focal MAGs most represented the orders Rhizobiales ($n = 8/40$), Actinomycetales ($n = 6/40$), Burkholderiales ($n = 6/40$), and Sphingomonadales ($n = 6/40$) (Supplementary Data 3).

Focal MAGs exhibited varied seasonal patterns in their overall read recruitment for both metagenome (Fig. 4A–C) and metatranscriptome (Figs. 4D, E, S2), with early, late, and consistent seasonal detection observed. However, in the metatranscriptomes of both years (for switchgrass), there was a general seasonal enrichment of particular KEGG subsystems represented by the transcripts over time, with increases in the later months of sampling (Fig. S3). Comparing the

 

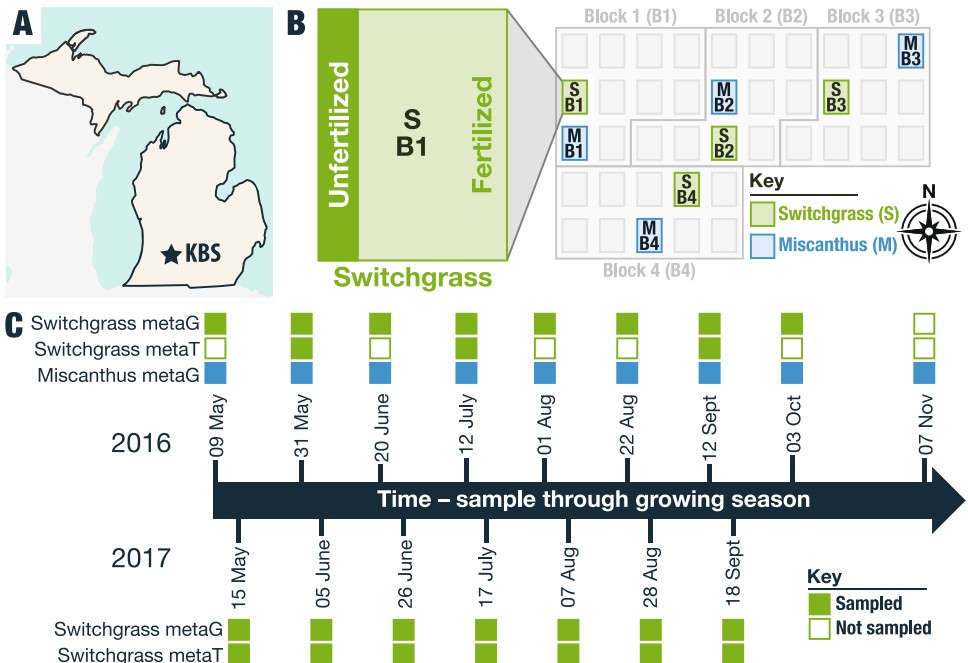

**Fig. 1 | Phyllosphere microbiome field sampling strategy at the Great Lakes Bioenergy Research Center Bioenergy Cropping System Experiment (BCSE) in 2016 and 2017. A** The study site is at Kellogg Biological Station (KBS), a Long-Term Ecological Research site focused on agroecosystems located in southwest Michigan. **B** Four replicate randomized cropping system blocks from the Biofuel Cropping System Experiment (BCSE) were sampled at each time point for switchgrass (green) and/or miscanthus (blue). Within each plot, a fertilized main plot and unfertilized subplot were sampled. **C** Symbol fill shows which sequencing was performed on samples from which time points, for which crop. Filled squares show samples that were collected and open squares show samples that were not collected at a particular time point, with a particular crop. In 2016, both switchgrass and miscanthus were sampled, and in 2017 only switchgrass was sampled. Switchgrass leaves were flash-frozen in liquid nitrogen for RNA extraction and metatranscriptome analysis at a subset of time points in 2016 and at all points in 2017. Notably, the new metagenome and metatranscriptome datasets presented here overlap with the samples of the 16S rRNA amplicon time series presented in Grady et al.[32] and the ITS amplicon time series presented in Bowsher et al.[96].

**Table 1 | Summary of RNA and DNA samples that returned reads and passed Illumina standard quality control at the Joint Genome Institute**

|  | 2016 | | 2017 | |
| --- | --- | --- | --- | --- |
|  | **Metagenome** | **Metatranscriptome** | **Metagenome** | **Metatranscriptome** |
| Switchgrass | 8 time points (64/64 successful) | 3 time points (22/24 successful) | 7 time points (56/56 successful) | 7 time points (56/56 successful) |
| Miscanthus | 9 time points (72/72 successful) | Not assessed | Not assessed | Not assessed |

The total collected samples submitted for sequencing is provided first, and the number of quality sequencing datasets returned for analysis is given in parentheses.

cumulative abundances of transcripts of each MAG between early (May – June) and late (July – September) 2017 sampling dates showed significant increases in transcription in 38 of 40 MAGs and decreased transcription in 1 MAG (S28) (two-sided Kruskal-Wallis test based on chi-squared distribution, $p$-value <0.05). Phenologically, August samples corresponded with (late) peak biomass and fruiting for switchgrass and with a closed canopy for both switchgrass and miscanthus. Senescence occurred as early as mid-September for switchgrass and as late as November for miscanthus. Notably, metatranscriptome read recruitment to the MAGs was robust even when there was inconsistent detection in metagenome read recruitment (Fig. 4), suggesting that these MAGs were consistently active and had relatively high activity even when their genome detection was obscured.

MAGs were clustered based on coherence in seasonal transcript dynamics (Figs. 4D, E, S2). Five coherent and statistically supported groups (clusters) of 40 MAGs were identified (Fig. S2). Cluster 1 contained a single MAG S28, assigned to genus *Pseudomonas* (S28, 98% complete, 1.9% contamination, 98.8% average nucleotide identity to *P. ceresai* GCA_900074915.1)[39] and its genes were enriched early in the season (2017), but the MAG was active throughout the season. The other four clusters contained numerous MAGs and were relatively

more dynamic, with trends towards late-season enrichment in transcripts. Several clusters contained MAGs annotated to the same class, suggestive of taxonomic coherence in seasonal activities. For example, cluster 3 included all but one (of eight) Actinomycetia MAGs.

## MAG genes and activities support a leaf-associated lifestyle

Not surprisingly, the most abundant subsystems identified among the 40 focal MAGs were associated with bacterial growth, such as carbohydrates, energy, amino acid, and nucleotide metabolisms (Fig. 5). Leaf metatranscriptomes also showed that most subsystem transcripts were steadily enriched over the season (15/21 subsystems were significantly increased; $p$-value <0.05, Table S2). Most transcripts increased over the season, with some exceptions, including those associated with colonization. A closer investigation of functions that trended to decrease over the season revealed that they could be attributed almost exclusively to MAG S28 (the only member in cluster 1 and most closely associated with *Pseudomodales*, Figs. S3 and S4). Finally, a few KEGG classifications were relatively stable and lacked overall significant seasonal trends, including transcripts for cell motility, antimicrobial resistance, adaptation, signal transduction, and translation.

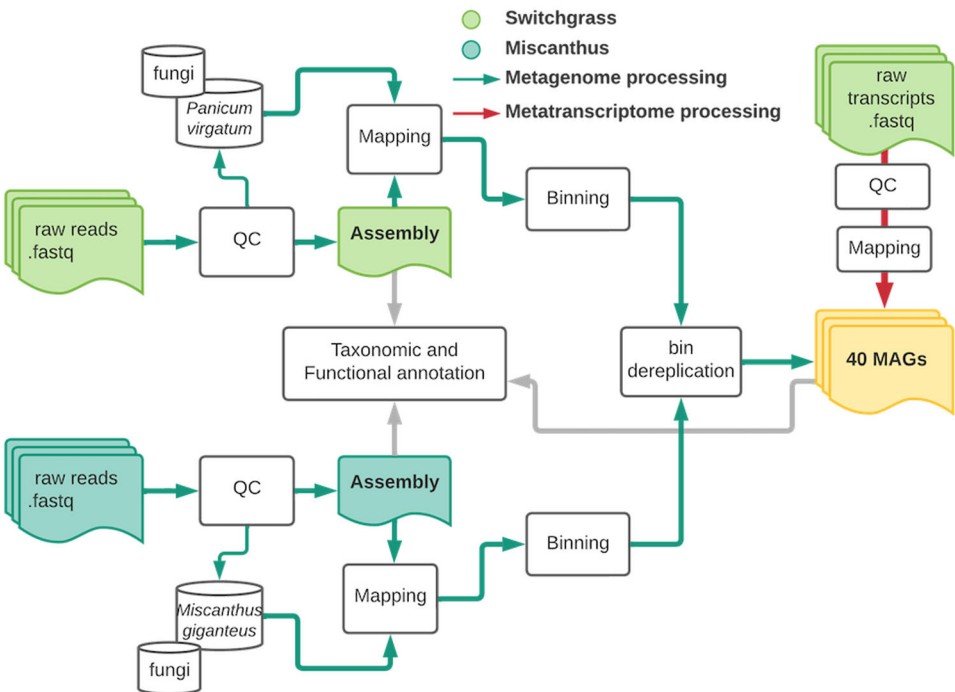

**Fig. 2 | Overview of bioinformatic processing of the metagenome (solid green arrows) and metatranscriptome (solid red arrows) datasets.** Switchgrass reads are shown in light green, and miscanthus is dark green. The solid grey arrow represents the common analysis step for both datasets. Figure was made with Lucidchart. Note that metatranscriptome data are from switchgrass only and not miscanthus.

A total of 124 distinct functional roles were identified in at least 39 of the 40 MAGs (Fig. S5). These functional roles were generally broad and associated with translation ($n = 48/124$), metabolism of energy ($n = 29/124$), and metabolism of carbohydrates ($n = 28/124$). There were also several functional roles associated with cofactors and vitamins ($n = 18/124$). While many functions were enriched in transcripts in the late season (July-Sept) relative to the season (May-June), three functional roles, in particular, stood out with enrichments greater than 20-fold in the late season transcripts. These three functional roles included proteins classified as short-chain dehydrogenase, molybdopterin oxidoreductase, and polyketide cyclase (Table 2, Fig. S5).

We then performed metabolic, biosynthetic, and plant-associated gene pathway analyses to understand the functions detected among focal MAGs in the phyllosphere and their activities inferred by transcript recruitment to the pathways (Fig. 6). We hypothesized that some of the pathways shared between bacterial populations may represent functions for fitness on the leaf. Pathways for terpene metabolism (34/40), betaine biosynthesis (30/40), trehalose metabolism (25/40), cyanide degradation (21/40), ROS degradation (23/40), and indole acetic acid (IAA) degradation (40/40) were the most common pathways shared and active across MAGs and were found in lineages from all four classes represented by the focal MAGs. Thus, these pathways likely are common among phyllosphere members and highly supportive of a leaf-associated lifestyle. In addition, there were some functional pathways and activities that may be more specialized functions because they were detected in only a few MAGs (e.g., xylitol metabolism). However, there was no clear phylogenetic pattern to the distributions of these putative specialized functions.

Because of the near-ubiquitous detection of terpene-related biosynthetic genes on the focal MAGs, we examined their annotations and recruitment more closely using gene function predictions (i.e., ORFs), which was expected to improve the sensitivity of functional annotation relative to metabolic pathway tools (e.g., BGC). We detected 278 ORFs associated with terpenes annotations in the metatranscripts. Collectively, these terpene-related genes followed the trend of gradual

enrichment over the season (Fig. 7A) and were relatively abundant in both years. The highest enrichment subcategory was terpenoid backbone biosynthesis, which included 29 ORFs. Among several isoprene biosynthesis genes detected, of interest was relatively the high and seasonal enrichment of the two terminal enzymes in the non-mevalonate isoprene biosynthesis pathway that is employed by bacteria, the *gcpE* and *lytB* (Figs. 7B, and S6). *GcpE* and *lytB* gene transcripts were detected in more than a third of the MAGs (13/40), and these included MAGs representing all three phyla detected. Six of the eight Proteobacteria that recruited isoprene biosynthesis transcripts were annotated to *Methylobacterium* (alpha). The four Actinomycetia were more distributed phylogenetically and annotated as genera *Frigoribacterium* M1, *Microbacterium* M105, *Amnibacterium* M67, and Pseudokineococcus M86. The single Bacteroidota that had isoprene biosynthesis transcripts was a *Hymenobacter* M9. We then investigated the 40 MAGs for detection of any of the nine genes previously reported to be involved in bacterial biosynthesis of isopentenyl diphosphate, a precursor to terpenoids like isoprene[40] and found that 29/40 MAGs had six or more of the genes detected and that all MAGs had at minimum 2 of the genes (Table S3). The consistent detection of genes associated with isoprene pathways in these MAGs, which are >50% complete, suggests that biosynthesis of isoprene-related molecules may be a prominent leaf strategy by phyllosphere bacteria. *Pseudomonas* MAG S28, noted previously to be the dominant population that colonized and activated early in the season (Figs. 4 and S2), had high isoprene biosynthesis transcript enrichment early in the season that then declined. However, the other eleven MAGs harboring genes from isoprene biosynthesis pathways then had increased activity in the late season.

Returning to the BGC annotations (by antiSMASH, Fig. 6), most of the terpene transcripts discovered through this approach were associated with pigment biosynthesis (e.g., carotenoids). The second most consistent class of BGC transcripts on the focal MAGs were non-ribosomal peptide synthase (NRPK) genes, but the majority of these did not have additional annotations beyond the general category.

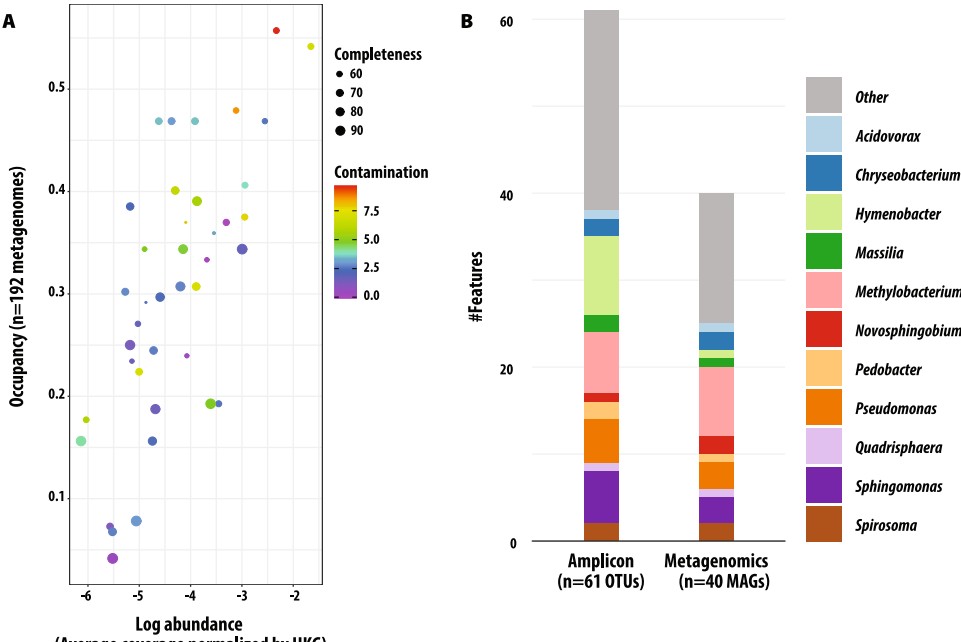

**Fig. 3 | Summary of leaf-associated MAGs. A** Abundance and occupancy of genomes assembled and binned from switchgrass and miscanthus phyllosphere metagenomes. Quality and contamination assessment were determined using checkM. 40 focal MAGs were selected. Occupancy is the proportion of the 192 metagenomes in which a MAG was detected, and abundance is the average normalized by housekeeping gene (HKG) MAG coverage across the 192 metagenomes. Symbol size shows the percent completeness of the MAG and color shows the percent contamination, with cooler colors indicating lower contamination. **B** Taxonomy of focal MAGs, as annotated with GTDB-tk, and taxonomic overlap with the persistent taxa detected in our previous 16S rRNA gene amplicon survey[32], which was conducted on the same samples and time series. Source data are provided as a Source Data file.

Therefore, the ORF analysis and BGC detection were complementary in results, especially for terpene- and isoprene-related functions. Another notable BGC finding was that all but two (of eight) Actinomycetia MAGs had transcripts for type III polyketide synthases, while this BGC class was less commonly detected among Proteobacteria and Bacteroidota.

### MAGs were detected in metagenomes from different field sites, crops, and years

We next asked if these focal MAGs were detected more broadly in other crop metagenomes. Because leaf metagenomes were not available, we obtained related soil metagenomes from local cropping systems in Michigan and also from the publicly available data within the Integrated Microbial Genomes (IMG) database, which serves as a repository for the Joint Genome Institute's sequencing efforts through the U.S. Department of Energy. We were surprised to find that, for each metagenome investigated, reads could be mapped to this set of MAGs. This included soil metagenomes from both switchgrass and miscanthus fields in three locations (Iowa, Michigan, and Wisconsin) and sampled across five different years (Fig. 8). These results suggest that the detected and analyzed MAGs can be widely distributed in midwestern agroecosystems and potentially of general importance for perennial crop environments.

## Discussion

Here, we report a multi-year seasonal metagenome and metatranscriptome assessment of plant phyllosphere in agricultural field conditions, focusing on the bacterial functions associated with two promising biofuel feedstocks. We expect these findings to have relevance for other grasses or systems with substantial aerial biomass, including native prairie. Furthermore, our collection of MAGs included phyllosphere members previously reported as abundant, persistent, or significant for microbiome assembly in other plants, including the model *Arabidopsis* (e.g.,[41], specifically: Sphingomonadales,

Pseudomonadales, Actinomycetales, Burkholderiales, and Rhizobiales. Therefore, the patterns and consistently detected functions among this curated collection offer general insights as well as seasonal phyllosphere functions.

There are multiple lines of evidence that the focal MAGs discussed here represent ecologically important lineages in the switchgrass and miscanthus phyllosphere. First, there is ample overlap among these focal MAGs and the core taxa of high abundance and occupancy from our previous 16S rRNA gene amplicon analysis of the switchgrass and miscanthus phyllosphere diversity[32], including a MAG associated with *Hymenobacter* M9, which is a genus that had high occupancy across fields and over time, as well as several OTUs assigned. This same time series was investigated in our previous amplicon analysis, and the core taxa were prioritized using consistency across replicated fields at the same time point, persistence over time, and relative abundance. This suggests that the populations represented by the MAGs are not rare taxa that are transient to the system. Second, we were able to generate quality assemblies from the complex metagenome data, which is a process that is generally biased towards abundant members. These MAGs are highly represented in read abundance in metagenomes and recruited relatively more metatranscriptome reads as well, suggesting that they are both abundant and active in the phyllosphere. Though it is possible that there are additional, ecologically important lineages missing from our collection, we are confident that those discussed here are among the major host- or environment-selected populations inhabiting the switchgrass and miscanthus phyllosphere.

### Stress responses: trehalose, betaine, reactive oxygen, IAA

Trehalose is a disaccharide that protects cells against salt, water, and osmolyte stress by serving as a stabilizing chemical chaperone, either by displacing water from protein surfaces or by vitrifying around protein structures to shield them[42]. Similarly, betaine is another commonly biosynthesized osmoprotectant used by microorganisms to contend with water, salt, and temperature stress[43]. Both have been

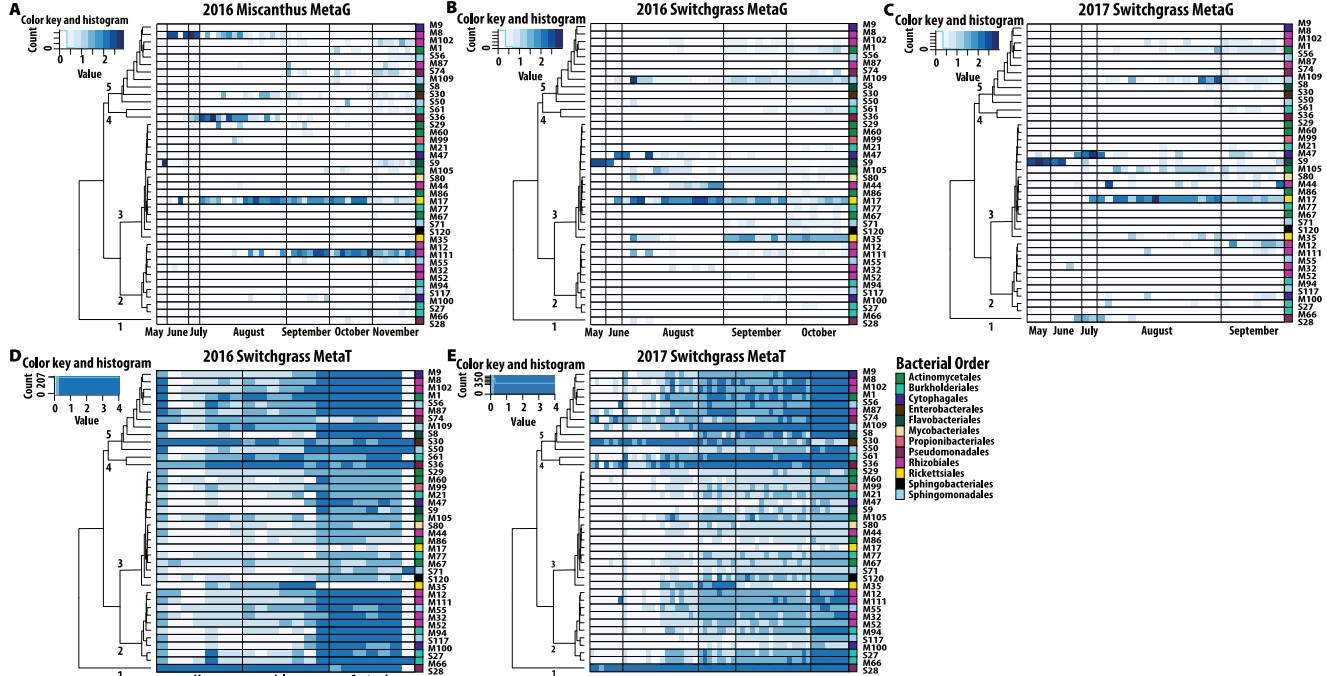

**Fig. 4 | Seasonal patterns of the 40 focal MAG metagenome and metatranscriptome read recruitment.** Abundance is indicated by color intensity, with blue indicating high and white indicating low abundance. MAG abundances for: **A** Miscanthus 2016 metagenomes (metaG); **B** Switchgrass 2016 metagenomes; **C** Switchgrass 2017 metagenomes; **D** Switchgrass 2016 metatranscriptomes (metaT) and **E** Switchgrass 2017 metatranscriptomes. Abundances of metagenome contigs were estimated with the median base pair of read recruitment divided by the average median base pair coverage of housekeeping genes. Abundances of metatranscriptome ORFs were estimated based on the median basepair coverage of all reads mapped to ORFs and divided by the median basepair coverage of housekeeping genes. The same dendrogram is applied to all panels, and it is the result of hierarchical clustering (see Fig. S2) of metatranscriptome diversity and abundances in switchgrass. Source data are provided as a Source Data file.

hypothesized to be important survival strategies of microorganisms in the phyllosphere, with supporting evidence from isolate genome analyses[44]. Here, we show both trehalose and betaine to be prominent among MAG populations and activated consistently on the leaf surface, suggesting that they are not seasonally activated but rather necessary for the leaf-associated lifestyle.

In bacteria, trehalose metabolism prevents cellular overflow metabolism and carbon stress by redirecting glucose-6-phosphate from conversion to pyruvate[42]. Trehalose biosynthesis is common among bacteria and archaea that live in arid, saline, thermal, or seasonally dry environments (e.g.,[45–47]) and it has also been reported to be induced in a pseudomonad by ethanol[48], which can originate inside plant cells[49] or more generally in the roots (e.g.,[50]), especially, during stress, fruit ripening, or senescence[51]. In switchgrass and plants in general, trehalose concentration is increased in response to drought conditions[52], and its precursor, trehalose-6-phosphate, induces senescence when carbon is readily available[53]. Furthermore, the *A. thaliana* phyllosphere member *Sphingomonas melonis* was reported to regulate trehalose biosynthesis during growth conditions that promoted mild stress[54]. Given this, it makes sense that the majority of MAGs had enrichment of transcripts related to trehalose metabolism, which would support a plant-associated lifestyle during drought and host senescence[54].

Betaine biosynthesis in bacteria often begins with the oxidation of choline, which is a part of plant tissues and can be transported into the cell[55]. Indeed, choline degradation was detected in 9/10 Gammaproteobacteria MAGs here (Fig. 6). Microbial-derived osmolytes such as betaine and trehalose have been suggested as targets for biotechnology development to support crop stress tolerance. However, plants can biosynthesize betaine and can be divided into groups of those that do and do not accumulate it in concentrations that are supportive of stress

tolerance[56]. For example, in switchgrass, the concentration of glycine betaine was not predictive of differences in drought tolerance among different genotypes, while trehalose was, along with abscisic acid, spermine, and fructose[52]. Furthermore, our data suggest no notable microbial limitations in the genetic potential or activation of betaine biosynthesis in the phyllosphere.

Reactive oxygen species (ROS) serve as signals for various developmental and cellular processes in plants, and here we detected active pathways for ROS degradation in the majority of focal MAGs. Though the precise mechanisms are unclear, homeostasis of ROS is expected to be involved in senescence[57], which is relevant to our study given that at least some to a majority of senescing plants were observed per plot in August and September sampling dates, respectively. Additionally, ROS accumulates in plants that are exposed to abiotic stress, to negative effects. ROS degradation is one of many functions phyllosphere microorganisms employ to contend with expected fluctuations in ROS on the leaf surface (though these fluxes are difficult to measure[57]). Previously, several genes relevant to oxidative stress response were differentially regulated in a wild-type phyllosphere bacteria *Sphingomonas melonis* strain Fr1 as compared to a knock-out mutant for regulation of general stress response when both were grown in a medium expected to induce low levels of stress[54]. Given that managing plant ROS is a target for reducing crop stress and regulating plant development[57,58], it is possible that manipulating microbial ROS degradation could be applied as one tool to achieve such efforts, but much more research is needed to understand the microbial-host interaction given ROS exposure or accumulation, and any possible ROS signaling between them.

IAA is a phytohormone produced by plants to regulate many processes in growth and stress response (e. g.,[59,60]). It is also made

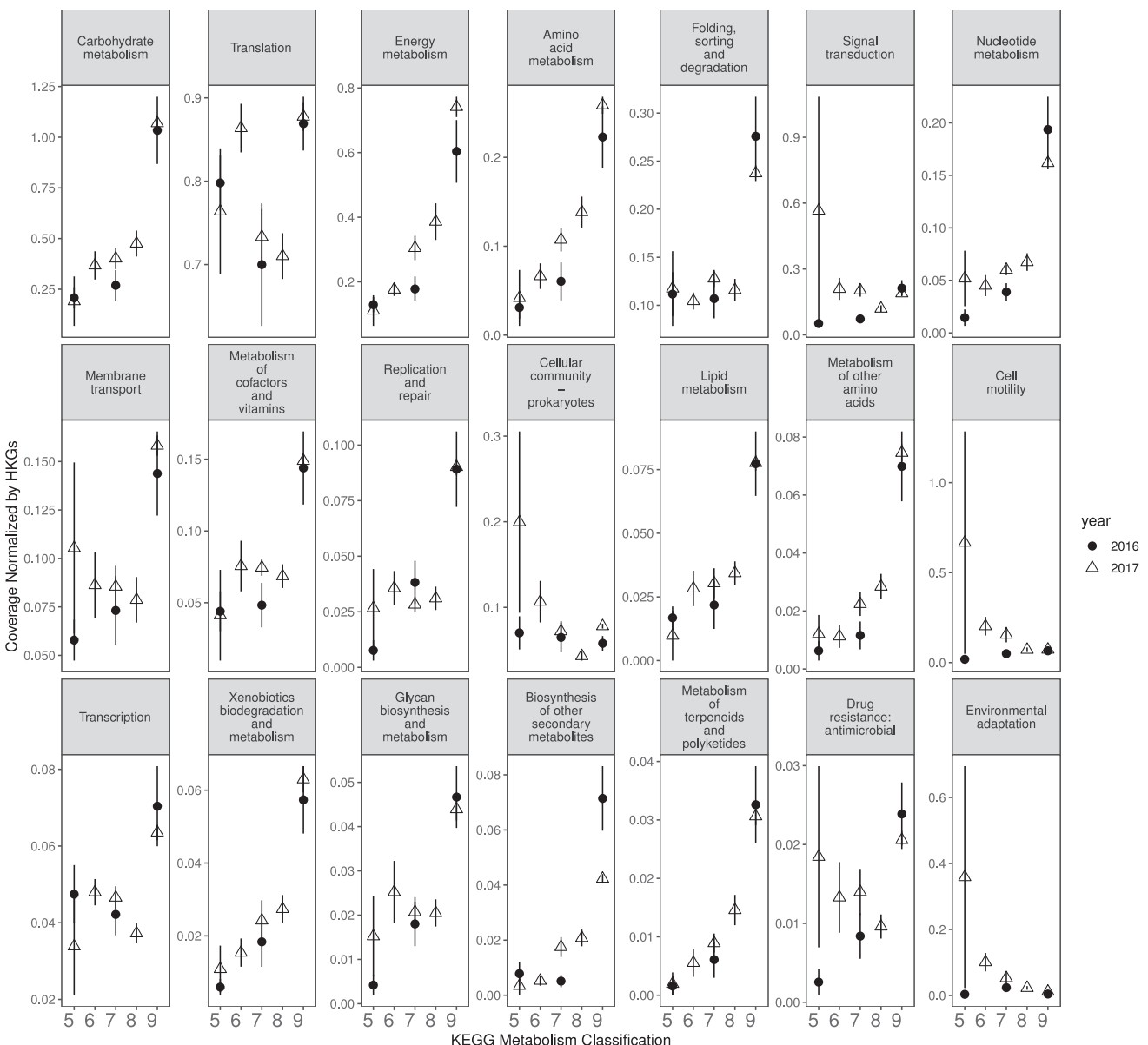

**Fig. 5 | 2016 (circle) and 2017 (triangle) switchgrass leaf transcript dynamics of KEGG metabolism classifications to the 40 focal MAGs.** The y-axis is scaled for each classification. Sample sizes are provided in Table 1 and included 22 and 56 metatranscriptomes for switchgrass in 2016 and 2017, respectively. Data are presented as mean values +/− standard error of the mean. Source data are provided as a Source Data file.

**Table 2 | Functional roles that exhibited strong seasonality with transcript enrichment by more than 20-fold in the late season than early**

| Functional role | Major classification(s) | Late: early ratio |
|---|---|---|
| Short chain dehydrogenase [PF00106.26]; Enoyl-(Acyl carrier protein) reductase [PF13561.7]; KR domain [PF08659.11] | Carbohydrate metabolism, Lipid metabolism, Nucleotide metabolism, Metabolism of cofactors and vitamins | 53.2 |
| Molybdopterin oxidoreductase [PF00384.23], Molydopterin dinucleotide binding domain [PF01568.22] | Carbohydrate metabolism, Metabolism of cofactors and vitamins, Energy metabolism, Signal transduction | 29.7 |
| Polyketide cyclase/dehydrase and lipid transport [PF10604.10] | Energy metabolism | 23.6 |

These roles were consistently detected among phyllosphere focal MAGs (39/40 detections).

by many microorganisms, including those shown to support plant growth promotion (e. g.,[60]). Therefore, the activity of IAA degradation pathways by focal MAGs is expected, given the redundancies between plants and microorganisms in synthesizing and responding to IAA, and demonstrates microbiome responsiveness to feedback in the host environment.

**Biosynthesis of isoprene-related molecules**

Most of the functions identified in our MAGs suggest general requirements for growth and maintenance given a leaf-associated lifestyle (e.g., carbohydrate and amino acid metabolism, pigment production to protect from radiation, similar to previous reports, e.g.,[61]. However, BGC analysis revealed surprising consistency in

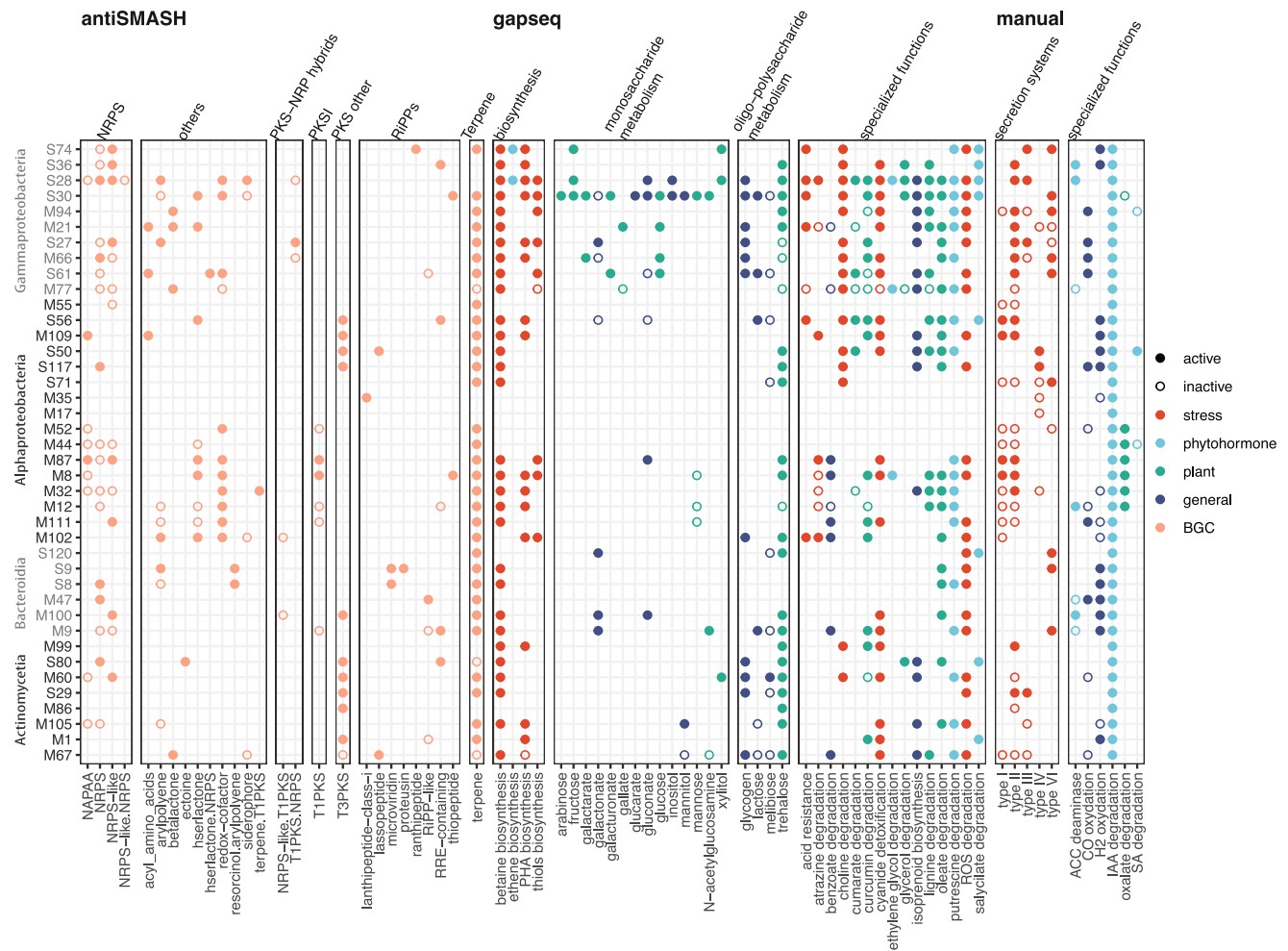

**Fig. 6 | Key functional gene pathways detected in the 40 focal MAGs and their activities (mapped transcripts) during the 2016–2017 switchgrass growing season.** Functional gene pathways were curated using antiSMASH for biosynthetic gene clusters, gapseq for general metabolic pathways, and manual selection for plant-associative functions reported in the literature. Pathways that were discovered in the MAGs but not detected in the transcripts are represented by open circles, and pathways detected in the MAGs and mapped by transcripts are represented by filled circles. Colors categorize different functional groups of pathways. Source data are provided as a Source Data file.

terpene metabolism pathways, leading us to look more closely at transcript ORFs associated with terpenes. This analysis revealed particular enrichment in pathways and genes associated with isoprenoid biosynthesis. Isoprenoids are a class of volatile terpenes that are generally abundant and reactive, and they engage in indirect and complex feedback with methane and nitrous oxide greenhouse gases[62]. Isoprene is one of the simplest isoprenoids. It is released by many plant species, and much of it is synthesized within the methylerythritol phosphate pathway of the chloroplast (MEP, aka:non-mevalonate)[63]. Isoprene is thought to act as a signaling molecule in stress response[64]. Studies have also found that isoprene emission protects leaf photosynthesis against short episodes of high temperature[65]. Plants emit isoprene from matured, photosynthetically active leaves, and emissions are light-responsive[63]. However, senescing leaves have been reported to decrease in their isoprene emissions relative to their leaves at peak growth[66]. Both switchgrass and miscanthus have been reported to emit relatively low basal levels of isoprene[67,68].

We speculate that members of the biofuel feedstock phyllosphere bacterial community may be either compensating for the loss of plant-derived isoprene, engaging in interspecies isoprenoid signaling with the host, protecting plant photosynthesis from thermal damage, quenching reactive oxygen species, or possibly producing isoprenoids as overflow metabolites (as hypothesized for *Bacillus subtilis*[69]). Bacterial isoprene degraders and synthesizers are widespread in nature[62] and have been previously investigated in phyllosphere communities of the relatively high isoprene emitter *Populus* spp[70], as well as in soils[71], which can serve as an isoprene sink. Stable isotope assays have been used to determine that a subset of bacteria community members degrade isoprene, including several Actinobacteria (*Rhodococcus* spp.) and *Variovorax* (Proteobacteria)[70,71]. Our MAG collection contains several Actinomycetia, a *Hymenobacter*, several *Methylobacterium*, and *Pseudomonas* MAG S28 that show activation of genes involved in isoprenoid biosynthesis and add support for their involvement in related molecular feedbacks in the phyllosphere. In addition, as these activities were detected in three Bacteria phyla, it suggests that biosynthesis of isoprene-related molecules may be a very common phyllosphere microbiome function. As isoprene is a precursor to sidechains needed for several quinones[72], it could be speculated that leaf bacteria scavenge isoprene emitted by the host plant to supplement bacterial synthesis of these sidechains but then compensate with de novo biosynthesis if the host decreases production. We observe that isoprenoid synthesis increases seasonally in the majority of MAGs containing these pathways and concurrently with when plant isoprene emissions also are expected to decrease, and directs future work to understand these dynamics and potential isoprenoid-mediated bacterial-host engagement.

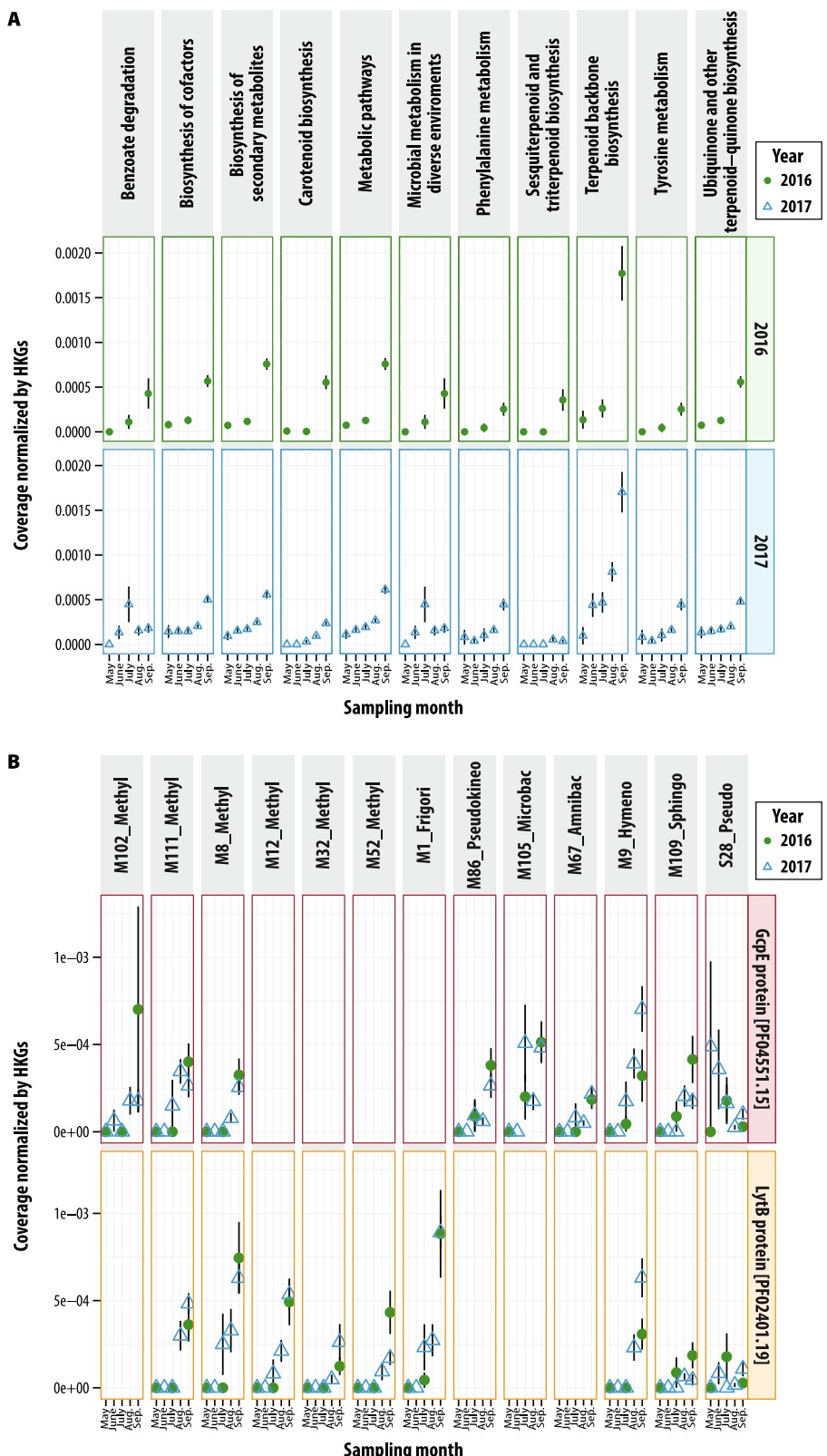

**Fig. 7 | Switchgrass microbiome transcripts detected over two growing seasons. A** 2016 (green, circle) and 2017 (blue, triangle) switchgrass leaf transcript dynamics of KEGG metabolism classifications associated with terpene metabolism. **B** Transcripts in MAGs associated with terminal enzymes in the non-mevalonate isoprene biosynthesis, *gcpE* and *lytB*. MAG IDs include predicted taxonomy at the genus level: Methyl = *Methylobacterium*, Frigor = *Frigoribacterium*, Pseudokineo =

*Pseudokineococcus*, Microbac = *Microbacterium*, Amnibac = *Amnibacterium*, Hymeno = *Hymenobacter*, Sphingo = *Sphingomonas*, and Pseudo = *Pseudomonas*. Sample sizes are provided in Table 1 and included 22 and 56 metatranscriptomes for switchgrass in 2016 (green circles) and 2017 (blue triangles), respectively. Data are presented as mean values ± standard error of the mean. Source data are provided as a Source Data file.

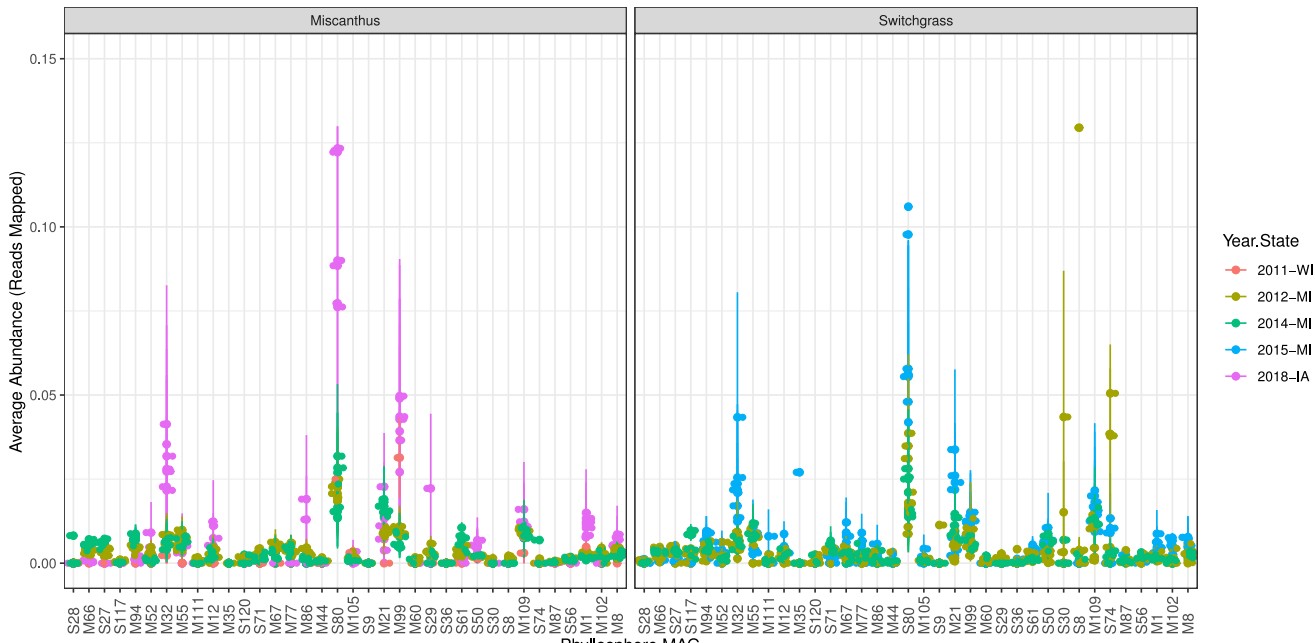

**Fig. 8 | Detection of 36 focal MAGs (out of 40) in publicly available metagen-omes of bioenergy grasses based on the average number of reads mapped to each MAG.** The sample size was 55 public metagenomes. Data are presented as mean values ± standard error of the mean. Source data are provided as a Source Data file.

## MAGs of interest

We highlight three MAG populations that were of interest because of their taxonomy, functions, dynamics, or occupancy. All were shared with taxa in our prior 16S rRNA survey, supporting their inclusion as part of the "core" set that was selected by abundance and occupancy. First, high-quality MAG S28 (>97% complete, <2% contamination) was a prominent pioneer and active colonizer of the leaf (Figs. 4, S2 Group 1). MAG S28 is related to *Pseudomonas cerasi*, a species reported to have phytopathogen relatives[73], but we did not note any disease symptoms on the leaves analyzed. This population had expected traits of a strong surface colonizer, including colonization, adaptation, and motility subsystems. It also had six pathways related to phytohormone responses (out of 7 total phytohormone pathways observed in these data), including activated ethene biosynthesis, ACC deaminase, and degradation of ethylene glycol, putrescine, salicylate, and IAA. These data suggest that S28 has several mechanisms to engage or respond to the host via phytohormones.

Next, MAG M9, identified as *Hymenobacter*, was of interest because it was associated with the most numerous taxonomic group detected in our prior 16S rRNA gene survey[32] and not among the most typically investigated phyllosphere lineages in the literature. While M9 populations were first detected early in the season, their transcripts were enriched in the late season (Figs. 4 and S2 Group 5). MAG M9 had detected and activated galactonate, N-acetylglucosamime, and lactose metabolisms, which were not common among the focal MAGs. It also had activated benzoate, curcumin, and putrescine degradation, as well as cyanide detoxification, type VI secretion, and dihydrogen oxidation. While M9 also had some pathways that were common among these MAGs (e.g., ROS and IAA degradation, terpene biosynthesis), its suite of more sparsely detected pathways and functions suggest a specialized role in the phyllosphere community. Notably, M9 has 65% completion and 0% detected contamination, suggesting more functional potential remains to be discovered for this and similar *Hymenobacter* lineages inhabiting the phyllosphere.

Finally, we selected a representative Actinomycetia MAG M60, a *Quadrisphaera* lineage that had activated isoprenoid biosynthesis and had increased activity late in the season, along with the majority of

focal MAGs (Figs. 4 and S2 Group 3). Studies have found that members of Actinomycetia are an important part of the phyllosphere that contribute to disease prevention and plant growth[74–76]. MAG M60 had several oligo/polysaccharide metabolisms that were infrequently detected in these data, including glycogen, melibiose, and trehalose. Despite its high completeness and low contamination (>95% and <5%, respectively), M60 was sparsely annotated by the methods we applied. However, *Quadrisphaera* taxa have been reported to be highly abundant in the phyllosphere or endosphere of various plants[77].

## Conclusions

Many recent review, perspective, and opinion pieces have urged integration of multi-omics approaches to improve understanding of the microbiome and its relationship to the host plant[44,78–81]. However, most integrative studies have focused almost exclusively on the rhizosphere as the compartment of soil-plant feedbacks and nutrient and water acquisition for the host. Though leaves are readily accessible for sampling, the phyllosphere microbiome is challenging to investigate using throughput, cultivation-independent approaches like metagenomics and metatranscriptomics. There are high levels of host and chloroplast contamination in leaf samples, and relatively low microbial biomass per leaf that that must be first dislodged from tightly-adhered biofilms. A signal from messenger RNA in metatranscriptome analysis is masked by abundant ribosomal RNA signal, leading to further challenges. These two challenges result in a relatively low proportion of usable sequences relative to the total sequencing effort (after quality filtering non-target signal) for leaf microbiome studies, which was true also in this work and is a limitation of it. Because of the combination of all of these challenges, much of our understanding of the phyllosphere, as the largest surface area of microbial habitation on Earth[26], has been learned from studies that employ model hosts and synthetic or model microbial communities in controlled settings, or from description of the community structure by sequencing of marker genes, amplified and bioinformatically depleted of chloroplast genes to overcome the challenges of low signal and host contamination.

Here, we report optimized laboratory protocols (to minimize host and chloroplast signals) combined with a genome-centric

bioinformatic approach to perform focused functional gene and transcript analysis of seasonally dynamic yet persistent phyllosphere microbiome members. Our work is an untargeted bacterial metatranscriptomic work performed on the leaf phyllosphere of field-grown crops. Other recent leaf metatranscriptome studies have focused the viral communities of tomato and pepper[82], soybean[83], and rice[84]. A key strength of this work is the challenging integration of phyllosphere metagenome and metatranscriptome data, leveraging the higher coverage of the metagenomes with the activity information available from the metatranscriptomes. Despite the relatively limited coverage of the MAGs (due to substantial host and ribosomal DNA contamination), the analysis proved successful by integrating both datasets and focusing on genome-centric interpretation. Thus, there are likely many more prevalent and functionally active populations of the phyllosphere that were not captured in this study, including those players known to be key in the phyllosphere (e.g.[20,85]). Substantial additional sequencing effort or an enrichment strategy would improve signal for a cultivation-independent approach to target those players. While the use of genome-centric approaches has the obvious shortcoming that we have obviously not captured every microbiome member, our approach does allow us to link actively transcribed functions to specific microbial membership. Furthermore, the functional genes and activities documented here are logical given current understanding of microbial adaptation to the host and phyllosphere environment.

Overall, this work provides evidence of a thriving, dynamic, functionally diverse, leaf-specialized, and host-responsive microbiome on the phyllosphere of perennial grasses. It provides evidence of specific phyllosphere functions that are seasonally activated in a temperate agroecosystem and suggests several hypotheses of important host-microbe interactions in the phyllosphere, for example via central metabolism, isoprenoid biosynthesis, and stress response engagements. This research contributes to our broad understanding of the dynamics and activities of phyllosphere microbial communities, and points to specific microbial functions to target that could prove useful for plant-microbiome management.

## Methods

### Site description and sampling scheme
Switchgrass and miscanthus leaves and corresponding contextual data were collected at the Great Lakes Bioenergy Research Center (GLBRC) located at the Kellogg Biological Station (KBS) in Hickory Corners, MI, USA (42°23′41.6″ N, 85°22′23.1″ W), at the GLBRC Biofuel Cropping System Experiment (BCSE) sites, within plot replicates 1–4 according to the site protocol, which involved walking along a transect and stopping at pre-determined sampling stations randomly situated within the fields so as to minimally disrupt the crop (Fig. 1). We sampled switchgrass (*Panicum virgatum L.* cultivar Cave-in-rock) and miscanthus (*Miscanthus x giganteus*) by collecting leaves at eight and nine time points, respectively, in 2016 and for switchgrass at seven time points in the 2017 season (Table 1, Fig. 1, Supplementary Data 1, Supplementary Data 2). We collected leaves for RNA isolation at three phenology-informed switchgrass time points in 2016 (emergence, peak growth, and senescence) according to GLBRC standard phenology methods [https://data.sustainability.glbrc.org/protocols/165] to assess the potential for sufficient mass and quality RNA extraction from the switchgrass leaf surface, and then expanded to include leaves from all switchgrass sampling time points in 2017. Leaves for RNA isolation were flash-frozen in liquid nitrogen immediately and stored at −80°C until processing.

### Phyllosphere RNA and DNA isolation
Phyllosphere epiphyte DNA was isolated and processed[32]. The same phyllosphere samples and extracted nucleic acids were used for this study as for Grady et al. 2019 (16S rRNA amplicon analysis) and for Bowsher et al. 2020 (ITS amplicon analysis). DNA concentrations were

normalized to 4 ng/ul. Phyllosphere epiphyte RNA was isolated using a benzyl chloride liquid: liquid extraction, based on[32] that was newly modified for RNA isolation based on the published methods of[86]. Approximately 5 g of intact, frozen leaf material was added to a 50 ml polypropylene conical tube (Corning #430290) and kept frozen on liquid nitrogen while samples were weighed and transferred. Denaturing Solution (D.S.) was prepared with 4.2 M guanidine thiocyanate, 25 mM sodium citrate dihydrate pH 7.0, 0.5% (v/v) sodium n-lauroyl sarcosine in Milli-q water and was filter sterilized through 0.22 mm filters. Immediately prior to extraction, a working stock of D.S. was prepared fresh by adding 2-mercaptoethanol to a final concentration of 5% (v/v, DS/2-ME). To each leaf tube, 5 ml of benzyl chloride, 2.5 ml of 3 M sodium acetate (pH 5.2), and 5 ml of the working stock of DS/2-ME were added. The tube was incubated in a 60 °C water bath for 20 min with vortexing every 1 min. The leaves were removed from the conical tube using ethanol-sterilized forceps and discarded.

Five ml of chloroform: isoamyl alcohol (24:1) were added to the remaining solution in each conical tube, which was then shaken by hand for 15 seconds and incubated on ice for 15 min. The tubes were then centrifuged at 12,000 × g for 15 min at 4 °C to separate aqueous and organic phases. Up to 5 ml of the upper aqueous phase was transferred to a clean 15 ml polypropylene tube without disrupting the white interface. Two and a half ml of sodium citrate buffer (1.2 M sodium chloride, 0.8 M sodium citrate dihydrate in Milli-q water, filter sterilized at 0.22 mm) and ice-cold isopropanol were added to achieve a final volume of 12.5 ml. Next, the tubes were centrifuged at 12,000 × g for 15 min to pellet the RNA, and the remaining supernatant was aspirated using a pipette. The RNA pellets were resuspended in 0.3 ml of working DS/2-ME solution, and afterward, 0.3 ml of ice-cold isopropanol was added and mixed by pipetting gently. The solution was incubated for 30 min at −20 °C, and then the full volume was transferred to a clean nuclease-free 1.7 ml tube and centrifuged at 16,000 × g for 15 min at 4 °C. The supernatant was removed, and the pellet was washed in 1 ml of nuclease-free 75% ethanol. Tubes were then centrifuged at 16,000 × g for 15 min at 4 °C, and the supernatant was again removed using a pipette. The remaining pellet was air dried to completely remove residual ethanol, then resuspended in 30 ml of nuclease-free Tris-EDTA buffer, pH 8.0.

Genomic DNA (gDNA) was removed using RNase-free DNase I (Thermo Fisher #AM2222) per the manufacturer's instructions. The RNA was then purified using the RNeasy MinElute Cleanup Kit (Qiagen Germantown, MD, USA) according to the manufacturer's instructions. The absence of contaminating gDNA was confirmed by the lack of amplification of the 16S rRNA gene V4 region by PCR[87] with positive and negative controls. This RNA isolation method was developed to most closely align with our established phyllosphere epiphyte DNA isolation[32] to minimize potential bias introduced during biofilm disruption or microbial cell lysis, as well as to minimize contamination from host RNA or genomic DNA by leaving the plant tissue intact. Commercial RNA extraction kits are primarily based on grinding or bead-beating whole tissue samples, which would result in increased over-representation of host-derived nucleic acids and potentially introduce bias in microbial cell lysis efficiencies.

### Metagenome and metatranscriptome library preparation
The Joint Genome Institute (JGI) performed the library preparation and sequencing from submitted RNA and DNA samples. Plate-based DNA library preparation for Illumina sequencing was performed on the PerkinElmer Sciclone NGS robotic liquid handling system using the Kapa Biosystems library preparation kit. 1.82 ng of sample DNA was sheared to 436 bp using a Covaris LE220 focused-ultrasonicator. The sheared DNA fragments were size selected by double-SPRI, and then the selected fragments were end-repaired, A-tailed, and ligated with Illumina compatible sequencing adaptors from IDT containing a unique molecular index barcode for each sample library. The prepared

libraries were quantified using KAPA Biosystems' next-generation sequencing library qPCR kit and run on a Roche LightCycler 480 real-time PCR instrument. Sequencing of the flowcell was performed on the Illumina HiSeq 2500 sequencer following a 2 × 151 indexed run recipe.

At JGI, plate-based RNA sample prep was performed on the PerkinElmer Sciclone NGS robotic liquid handling system using Illumina Ribo-Zero rRNA Removal Kit (Bacteria) and the TruSeq Stranded Total RNA HT sample prep kit following the protocol outlined by Illumina in their user guide: https://support.illumina.com/sequencing/sequencing_kits/truseqstranded-total-rna.html, and with the following conditions: total RNA starting material of 100 ng per sample and ten cycles of PCR for library amplification. The prepared libraries were quantified using KAPA Biosystems' next-generation sequencing library qPCR kit and run on a Roche LightCycler 480 real-time PCR instrument. Sequencing of the flowcell was performed using a low input RNASeq protocol with rRNA depletion on the Illumina NovaSeq 6000 sequencer with NovaSeq XP V1 reagent kits, S4 flow cell, following a 2 × 151 indexed run recipe.

## Quality filtering of metagenomes and metatranscriptomes

We proceeded with bioinformatic analysis (Fig. 2) of 192 metagenome (Supplementary Data 1) and 78 metatranscriptome (Supplementary Data 2) observations that met JGI standards for raw data quality based on the Illumina proprietary software. We used Trimmomatic (v0.39)[88] to remove adaptors and filter low-quality reads from fastq files using the following arguments: P.E. -phred33 ILLUMINACLIP:TruSeq3-PE-2.fa:2:30:10:8:TRUE LEADING:3 TRAILING:3 SLIDINGWINDOW:4:15 MINLEN:36. After assembly, plant host reads were filtered out (for both metagenomes and metatranscriptomes) by removing all reads that mapped to the switchgrass genome (*Panicum virgatum* v1.0, DOE-JGI, http://phytozome.jgi.doe.gov/) or miscanthus genome (*Miscanthus sinensis* V7.1 http://phytozome.jgi.doe.gov/) using bowtie2 (v 2.4.1), samtools (v 1.13), and bedtools (v2.30.0)[89–91] (Fig. S1). To remove the fungal reads and improve the prokaryotic signal in metagenome samples, we also filtered reads against seven fungal genomes[92–95] that represent close relatives of the most abundant fungal species in this system that we assessed and reported in our prior work[96] (Table S1). The genomes of these fungal species were retrieved from the JGI Genome Portal or GenBank.

## Metagenome assemblies and metagenome-assembled genome binning, refinement, and annotation

Two metagenome assemblies, one for switchgrass and one for miscanthus, were created based on metagenomes collected in 2016. These filtered metagenome reads were combined and used for co-assembly (co-assembled by crop in 2016) with MEGAHIT (v 1.2.9) using --kmin-1pass (low sequencing depth) and --presets meta-large (complex metagenome)[97]. Additionally, we recovered metagenome-assembled genomes (MAGs) from the 2016 switchgrass and miscanthus metagenome libraries (*n* = 136 metagenomes) using Metabat (v2.2.15)[98]. MAG assemblies were performed using filtered reads from switchgrass and miscanthus sampled from 2016 separately to maximize completeness and reliability, as also done in other studies[99]. To assess the MAGs quality and completeness, we used CheckM (v.1.13 with the lineage_wf option)[100]. Among a total of 238 MAGs assembled from the switchgrass or miscanthus phyllosphere metagenomes (Supplementary Data 3), we selected a subset of MAGs based on: completeness greater than 50% and contamination less than 10%. We identified replicated bins associated with MAGs using dRep (v3.2.0, Olm et al. 2017), resulting in the removal of a single MAG. An additional MAG was removed due to insufficient read recruitment in metatranscriptomes (described below) for a final total of 40 focal MAGs, including seven high-quality and 33 medium-quality[38] (Supplementary Data 3).

Read recruitment was performed to the 40 focal MAGs[38,101]. The metagenome abundance of contigs in MAGs in each sample was estimated based on the median coverage of filtered metagenome reads associated with each MAG contig. Specifically, Bowtie2 (v2.2.2) was used to align reads to all focal MAG contigs (using the default setting and allowing for a single read to map to only one reference). Bedtools (v2.28) was used to estimate the coverage of each base pair within the contig. The estimated abundance of a contig was based on the median basepair coverage of all reads mapped to the contig, and the estimated abundance of a MAG was based on the average median coverage of all contig within its bins (Supplementary Data 4). The metatranscriptome abundance was estimated based on protein-encoding genes identified in MAGs. For each MAG contig, open reading frames (ORFs) and functional genes were identified using Prodigal (v2.6.3, default parameters). Transcripts were mapped to ORFs associated with each MAG to estimate the median base coverage of each ORF (Bowtie2, default parameters, no multiple mappings allowed). To normalize varying sequencing depths, we estimated abundances by normalizing them by the sum of the median base pair coverage of housekeeping genes in the focal MAGs identified in each sample (Supplementary Data 5). Housekeeping genes were identified based on full sequence alignment to the hidden Markov models (HMMs) of 71 housekeeping single-copy genes with an E-value of less than 1e-5 as standard[102,103]. If housekeeping genes were not identified in a metagenome or metatranscriptome, samples were removed from further analysis. We also estimated the total reads associated with each MAG for metagenomes and metatranscriptomes (Supplementary Data 6) (One MAG that had originally met completeness and contamination criteria, M22, had an average of 48 reads map to metatranscriptome and was then removed from further analysis.) For metatranscriptome analyses, only ORFs with the top 75% of observed abundances and detected in at least 10% of samples were considered.

Functional annotation of ORFs in focal MAGs was done with the DRAM tool (v1.1.1[104], using UniRef90, MEROPS, PFAM, dbCAN-HMMdb (v8) databases (all compiled with DRAM on February 12, 2021), and the KEGG database which we manually added to the DRAM pipeline (release January 1, 2018). To obtain functions related to terpene metabolism, ORFs were selected that were associated with any KEGG annotation that contained the phrase "terpen."

MAGs were assigned taxonomy using GTDB-tk (v1.4.0,[105]). Assembled focal MAGs were aligned against the chloroplast genome of *Panicum virgatum* (NC015990) (BLAST, 2.10.1). We detected partial matches to eight bins and no full alignments, confirming that those bins were not chloroplast genomes. Additionally, we compared the focal MAGs taxonomy to our previously detected 16S rRNA gene core cohort[32]. This cohort consists of 61 phylogenetically diverse bacterial OTUs (97% clustering). The taxonomy of these 61 OTUs was compared with the 40 MAGs at the genus level. Due to differences in nomenclature between the GTDB and SILVA databases, we removed extensions for MAGs classified as *Pseudomonas_E* or *Aeromonas_A*.

## Statistical analyses

Statistical analysis was performed in the R environment for statistical computing (inclusive of releases between 2021 and 2023). Increasing and decreasing seasonal trends in functional roles were assessed with linear regression. Pairwise comparisons of functional roles (the cumulative sum of all associated ORFs) between the early (May – June) and late (July – Sep) seasons were performed with the two-sided Kruskal-Wallis test based on the chi-squared distribution by ranks, with early and late seasons were delineated by plant phenology (flowering/senescence in late season). The distribution and dynamics of MAGs' transcripts were compared with Ward's method for hierarchical clustering using euclidian distances of the estimated metatranscriptome abundances using *pvclust* (version 2.20)[106].

## Assessment of biosynthetic gene clusters (BGC) on focal MAGs

Biosynthetic gene clusters (BGC) were predicted by antiSMASH (v6.0)[107] and further annotated by Big-SCAPE (v1.1.0)[108]. While

there are only 8 BGC classes used in Big-SCAPE (i.e., PKS I, PKS other, NRPS, RiPPs, Saccharides, Terpenes, PKS/NRPS Hybrids, and Others), antiSMASH provides a more detailed classification. We leveraged both outputs to investigate the diversity and expression of predicted BGCs in focal MAGs. From each predicted gene cluster, we extracted the location of the biosynthetic gene (i.e., gene_kind = "biosynthetic" and core_position). To evaluate the transcription of putative BGCs within a MAG, we searched for any transcripts that are mapped to the same genomic region of predicted biosynthetic genes.

### Predicting the functional potential of focal MAGs

We used gapseq[109] to predict completed metabolic pathways in our focal MAGs. We used the 'find -p all −b 200' option to search for pathways against the MetaCyc database. We filtered out incomplete pathways, and the remaining pathways were grouped into broader categories using MetaCyc classification and manually curated to focus on understanding the pathways relevant either for the plant environment or for microbial interactions with plants. These categories were defined as potential involvement in: i) plant (using plant metabolites/cell components), ii) phytohormone (known/potential involvement in phytohormone homeostasis), iii) stress (e.g., drought, reactive oxygen species), and iv) general (pathways that utilize potential plant-derived products). Furthermore, we also manually searched for genes/ pathways that were missed by gapseq but are known to be related to adaptation to plant-associated lifestyle, including secretion systems[110,111], oxidation of trace gases ($H_2$ and C.O.)[112,113], oxalate degradation, and phytohormone production/degradation. Similar to our biosynthetic gene analysis, the activity of predicted pathways was estimated based on mapped transcripts to identified genes. Contigs of focal MAGS were aligned (BLAST v2.7.1+) against nine isoprenoid precursor biosynthesis genes previously reported[40]. The top hit of each query was considered and accepted as aligned if E-value was less than 1e-5.

### Detection of MAGS in public metagenomes

To evaluate the presence of MAGs in other switchgrass and miscanthus-associated metagenomes collected from the field, we used 55 public switchgrass, miscanthus, and corn soil metagenomes (Supplementary Data 7). Metagenome reads were mapped to focal MAGs using the same methods for mapping and abundance estimation described above for the metagenomes generated in this study.

### Reporting summary

Further information on research design is available in the Nature Portfolio Reporting Summary linked to this article.

## Data availability

The raw and processed metagenome and metatranscriptome data generated in this study have been deposited in the Joint Genome Institute Genome Portal database under proposal ID 503249 [https://genome.jgi.doe.gov/portal/Seadynanfunction/Seadynanfunction.info.html]. The MAGs data generated in this study have been deposited in NCBI under bioproject PRJNA800073. The metadata, including metadata standards for metagenomes, metatranscriptomes, and metagenome-assembled genomes, for this study are provided in the Supplementary Data 1–7. Source data are provided with this paper.

## Code availability

Annotated code and links to data and metadata are available on GitHub (https://github.com/ShadeLab/PAPER_Howe_2021_switchgrass_MetaT) and Zenodo (https://zenodo.org/record/10040).

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

## Acknowledgements

Support for this research was provided by the Great Lakes Bioenergy Research Center, U.S. Department of Energy, Office of Science, Office of Biological and Environmental Research (Awards DE-SC0018409 and DE-FC02-07ER64494), by the National Science Foundation Long-term Ecological Research Program (DEB 1637653 and 1832042) at the Kellogg Biological Station, Michigan State University AgBioResearch, and by the DOE Center for Advanced Bioenergy and Bioproducts Innovation (U.S. Department of Energy, Office of Science, Office of Biological and Environmental Research under award number DE-SC0018420). This work was also supported in part by Michigan State University through computational resources provided by the Institute for Cyber-Enabled Research and in part by the University of Wisconsin-Madison Wisconsin Energy Institute as supported by GLBRC Information Services. The work [proposal:10.46936/10.25585/60000818] conducted by the U.S. Department of Energy Joint Genome Institute [https://ror.org/04xm1d337], a DOE Office of Science User Facility, is supported by the Office of Science of the U.S. Department of Energy operated under Contract No. DE-AC02-05CH11231. AS acknowledges support from the USDA National Institute of Food and Agriculture and Michigan State University AgBioResearch. N.S. acknowledges support from the Michigan State University Plant Resilience Institute.

## Author contributions

K.L.G. and A.S. conceived and designed experiments; N.S., K.L.G., and A.S. performed the experiments; A.H., N.S., S.K.D., F.Y., and A.S. analyzed the data; A.H., N.S., S.K.D., F.Y., K.L.G., and A.S. contributed materials/analysis tools; and A.H., N.S., S.K.D., F.Y., K.L.G., and A.S. wrote the paper.

## Competing interests

The authors declare no competing interests.
