## [Peer Review File · Nature Communications]

Seasonal activities of the phyllosphere microbiome of perennial cropsEditorial Note: This manuscript has been previously reviewed at another journal that is not operating a transparent peer review scheme. This document only contains reviewer comments and rebuttal letters for versions considered at *Nature Communications*. Mentions of the other journal have been redacted.

Reviewer #3 (Remarks to the Author):

I have reviewed this manuscript in [redacted]. I really appreciate the efforts and improvement from the authors, but it is still difficult to make solid conclusions at the metagenomic level. The intrinsic difficulty of high amount of contamination of plant DNAs resulted in very low sequencing depth of leaf microbiome samples. This is a general problem of leaf microbiome samples, but really undermine the power of metagenomic method. The 40 MAGs (the number is quite low) from 192 samples reveal the lack of sufficient abundant microbial reads. The authors did excellent work in taxonomic assembly of leaf microbiome using amplicon sequencing method in their previous work, but more reads from metagenomic data are needed anyway to reveal patterns genetic patterns of leaf microbiome.

Response to reviewers' comment

Reviewer #3 (Remarks to the Author):

I have reviewed this manuscript in [redacted]. I really appreciate the efforts and improvement from the authors, but it is still difficult to make solid conclusions at the metagenomic level. The intrinsic difficulty of high amount of contamination of plant DNAs resulted in very low sequencing depth of leaf microbiome samples. This is a general problem of leaf microbiome samples, but really undermine the power of metagenomic method. The 40 MAGs (the number is quite low) from 192 samples reveal the lack of sufficient abundant microbial reads. The authors did excellent work in taxonomic assembly of leaf microbiome using amplicon sequencing method in their previous work, but more reads from metagenomic data are needed anyway to reveal patterns genetic patterns of leaf microbiome.

>>>Thank you for this comment. We completely agree that there is an intrinsic methodological difficulty in metagenome analysis from plant tissues. We discuss the particular details of this (including high contamination of plant DNA, low biomass system, etc) in the discussion from Lines 430-457, Line 438 also added clear, explicit statement that this study also suffers from these limitations like the others (Line 443) and that additional sequencing improves observational effort (Line 457).